# Effect of Hybrid Laser Arc Welding on the Microstructure and Mechanical and Fracture Properties of 316L Sheet Welded Joints

**Linyi Xie** [1,2], **Wenqing Shi** [1,*], **Teng Wu** [1], **Meimei Gong** [1], **Detao Cai** [2], **Shanguo Han** [2] **and Kuanfang He** [3]

1   School of Electronic and Information Engineering, Guangdong Ocean University, Zhanjiang 524088, China
2   Guangdong Provincial Key Laboratory of Advanced Welding Technology, China-Ukraine Institute of Welding, Guangdong Academy of Sciences, Guangzhou 510650, China
3   School of Mechatronic Engineering and Automation, Foshan University, Foshan 528000, China
*   Correspondence: swqafj@163.com

**Abstract:** To explore the influence of different welding modes on the properties of 316L thin-plate welded joints, a new type of laser arc compound gun head similar to a coaxial one was used in this experiment. A high-speed camera was used to record the welding process and analyze the droplet splash behavior of the molten pool. The microstructure, microhardness change, and tensile test results of welded joints under different welding modes were analyzed. The results showed that laser welding (LW) is more prone to molten pool splash than hybrid laser arc welding (HLAW). The HLAW pool area was significantly increased compared with that of LW. The HLAW joint microstructure was more uniform than that of LW, which can improve the microhardness of welded joints. HLAW improved the tensile properties of the joint, with the maximum tensile strength of the joint increasing from 433 to 533 MPa. This test can provide guidance for the HLAW process.

**Keywords:** hybrid laser arc welding; the molten pool splashes; microstructure; tensile strength



## 1. Introduction

316L stainless steel has a low price, is corrosion resistant, and has high-temperature resistance along with other characteristics [1,2]. It is widely used in the petroleum, chemical, and other industrial fields [3]. Welding is utilized extensively as the connection between metal materials in industrial production [4]. Laser welding (LW) is more and more widely used in many industries, mainly because of its fast welding speed, good joint quality, and small heat affected zone [5–7]. However, the high cooling rate during LW leads to hardening of the microstructure and the heat affected zone of the weld [8,9]. How to limit the high cooling rate caused by LW has become a popular research topic [10–12].

In recent years, hybrid laser arc welding (HLAW) has attracted particular attention [13,14]. HLAW combines a laser beam with arc welding [15]. The high cooling rate caused by lasers can be mitigated using the arc, which can produce high-quality joints through a combination of the laser and arc [16]. Compared with LW, HLAW has many advantages even at higher welding speed [17,18]. HLAW can be subdivided into several specific welding methods based on laser (different wavelengths) and arc (TIG, MIG, plasma) classification. In the beginning, HLAW was proposed to improve the energy efficiency of laser ($CO_2$). The laser energy is transferred to the arc plasma, which leads to a heat conduction dominated by the arc. Each HLAW has obvious advantages in specific applications.

At present, HLAW is primarily based on the offset side shaft; that is, the laser and arc interact at a certain angle [19,20]. Chen et al. [21] conducted experimental research and thermoplastic analysis on multilayer bias paraxial HLAW of 316L stainless steel. They proposed reasonable heat sources for simulating HLAW and laser beam welding. Derakhshan et al. [22] conducted a comparative study of bias paraxial HLAW and conventional arc welding and found that a lower heat input had a significant effect on the final

deformation of the welded structure. Mu et al. [23] studied the metal vapor cooling effect of biased paraxial HLAW, broadened the interaction theory between the laser and plasma in HLAW, and promoted the development of HLAW.

With the continuous development of HLAW, research on coaxial HLAW has gradually matured. However, because of the difficulty in the integrated manufacturing of coaxial nozzles, coaxial HLAW welding has not been widely used. Lei et al. [24] conducted welding experiments on a self-designed laser hollow tungsten coaxial welding platform. The change in the arc physical field after laser addition was analyzed by numerical simulation, and the mechanism of improving welding depth was discussed. The main reasons for the increase in penetration depth of composite welding were found to be the increase in arc temperature, arc pressure, and thermal efficiency.

According to the above research, a new type of laser arc composite gun head similar to a coaxial one was used in this experiment. Different from conventional HLAW (1030 nm laser-TIG), which uses arc (over 100 A) as the main heat source, low-current arc (about 20 A) is used as the auxiliary heat source in this paper. To explore the role of small and medium current arc in HLAW, the properties of HLAW and LW welded joints were compared. A high-speed camera was used to record the welding process, and the results were used to analyze the splash behavior of molten pool droplets. The quality, microstructure, and mechanical properties of welded joints under different welding modes were compared. These results can provide some useful guidance for coaxial HLAW.

## 2. Experimental Method

The welding experiment in this study was completed on a self-built HLAW system, which is shown in Figure 1a. In the internal structure of gun head, the angle between the laser and the tungsten axis is 30° in Figure 1b. The arc is generated by the tungsten electrode, and it interacts with the laser from the same nozzle of the gun head. We used the following equipment in our experiments: a laser source, an arc plasma machine, a robot, a high-speed camera, and an infrared laser assisted light source. The laser source was provided by a 10-kW laser (TRUMPF TruDisk Ditzingen, Germany) at 1030 nm. The arc was generated by using a self-built control cabinet (PLAZER MP-1001-50). The welding trajectory motion was completed by using the si*x*-axis mechanical arm of a KUKA robot. A high-speed camera (pco.dimax HS2) was used to record the welding process video. An 808-nm filter lens was used to suppress the collection of arc plasma plume radiation and metal vapor, and an 808-nm infrared laser-assisted light source was used for backlighting [25,26]. The video was recorded at 3000 frames/s with a resolution of 748 × 792 pixels.

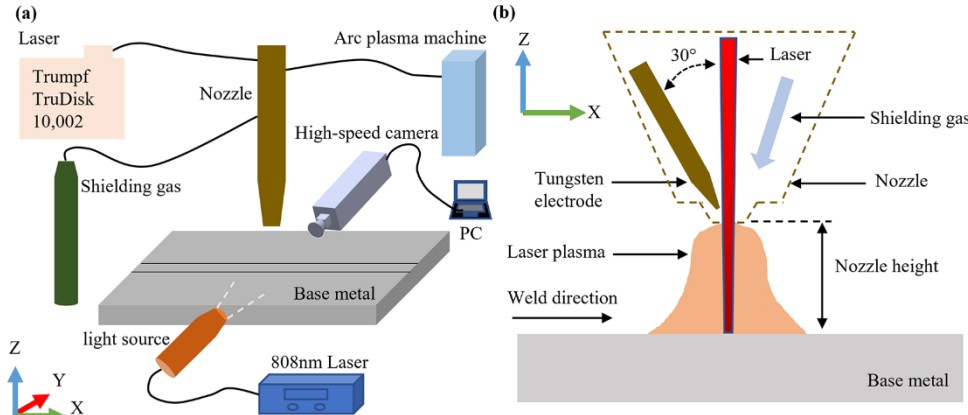

**Figure 1.** Schematic diagram of HLAW system: (**a**) HLAW equipment; (**b**) composite mode of laser and arc.

A 100 mm × 50 mm × 1 mm 316L stainless steel plate was used as the welding material, and its material composition is shown in Table 1. Argon was selected as the protective gas, and the airflow velocity was 10 L/min. The nozzle height was 3 mm, the

welding speed was 0.02 mm/s, and the arc voltage was 20–23 V. According to our previous study [27], the specific welding experimental parameters, as listed in Table 2, were all based on all well-formed considerations. After welding, the metallographic specimens were cut along the vertical direction of welding for observation. Then they were corroded with a corrosive solution (10 mL HNO3, 20 mL HCl, and 70 mL water). The joint was observed using a Carl Zeiss metallographic microscope (Axio lmager M2m). The microhardness of the joint was measured with a Wilson Vickers microhardness tester (Buehler VH1202), and the test position is shown in Figure 2a. Three standard tensile specimens were prepared for each experimental parameter, and the detailed size of the tensile specimen is shown in Figure 2b. The tensile test was performed on a universal tensile machine (MTS5105) at a constant tension of 1 mm/min. Tensile fracture was observed using a scanning electron microscope (FEI Quanta 250).

**Table 1.** Mass fraction of 316L (wt %).

| Type | Fe | C/% | Si/% | Mn/% | P/% | S/% | Mo/% | Cr/% | Ni/% |
|------|------|------|------|------|------|------|------|------|------|
| 316L | Balance | 0.027 | 0.62 | 1.06 | 0.042 | 0.004 | 2.14 | 16.12 | 10.02 |

**Table 2.** Test parameters for LW and HLAW.

| NO. | 1 | 2 | 3 | 4 | 5 | 6 |
|-----|------|------|------|------|------|------|
| Laser power (W) | 600 | 800 | 1000 | 600 | 800 | 1000 |
| Current (A) | 0 | 0 | 0 | 20 | 20 | 20 |

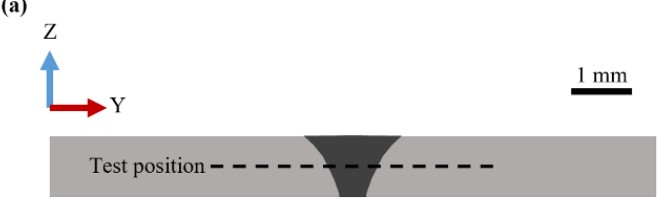

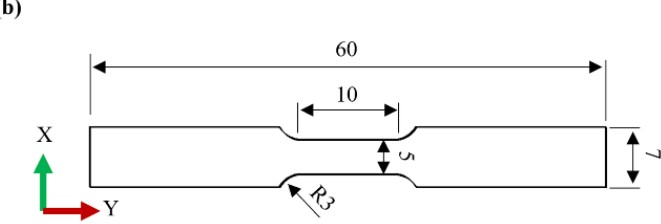

**Figure 2.** Microhardness test position and tensile size: (**a**) microhardness test position; (**b**) size of tensile (in mm).

## 3. Results and Discussion

### 3.1. High-Speed Camera Images

The welding process under different parameters was recorded using a high-speed camera. Figure 3 shows a single frame of the video, which was processed in pco.camware 64 to select the most informative regions. Figure 3a–c show images of the welding process under LW 600, LW 800, and LW 1000 parameters, respectively. Splash droplets in the molten pool are marked by a white coil. The central bright area is the molten pool area circled in red. Findings show splashing in LW and that the surface of the molten pool obviously fluctuates (Figure 3a–c). Figure 3d–f show images of the welding process under HLAW 600-20, HLAW 800-20, and HLAW 1000-20 parameters, respectively. The bright area illuminated from the top is an arc plasma partially seen through a filter lens. No splashes

are evident in Figure 3d–f; however, there are spatters on the surface of the base material (BM). In other words, both HLAW and LW showed molten pool splashing.

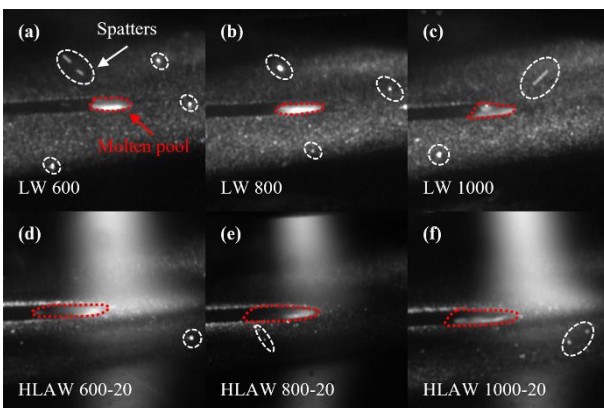

**Figure 3.** Static picture of welding process under different welding methods: (**a**–**c**) LW 600, LW 800, and LW 1000, respectively. (**d**–**f**) HLAW 600-20, HLAW 800-20, and HLAW 1000-20, respectively.

Figure 4 shows the dynamic process under different welding modes. A red ring is used to mark the weld pool area. Figure 4a–d show images under LW 600 at different times, respectively. The boundary and shape of the weld pool can be obviously seen in this area. The molten pool fluctuates violently within 9 ms. Figure 4e–h show images under HLAW 600-20 at different times. It can be observed that the surface of the molten pool was relatively stable at 0, 3, 6, and 9 ms.

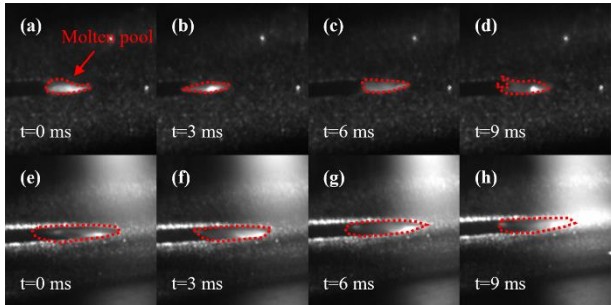

**Figure 4.** Dynamic process under different welding modes: (**a**–**d**) LW 600; (**e**–**h**) HLAW 600-20.

To further compare the splash of LW and HLAW in molten pool during the welding process, we calculated the molten pool area of each test parameter and the dynamic change process of the molten pool area by the ImageJ software in Figure 5. Figure 5a shows that the molten pool area of HLAW is about twice that of LW under the same laser power. This phenomenon is related to the arc energy distribution in HLAW. Because the heat affected zone of the arc is larger and the energy distribution is more dispersive than the laser. Thus, the molten pool area of HLAW is larger than that of LW. In Figure 5b, the dynamic changes in the molten pool area of LW and HLAW at different times can be clearly observed. In general, the range is the maximum and minimum difference. In this paper, the range of molten pool area reflects the fluctuation of the molten pool to some extent. The range of LW is 4.61 mm$^2$ and the range of HLAW is 3.90 mm$^2$. Therefore, molten pool fluctuation under LW is more violent than that under HLAW.

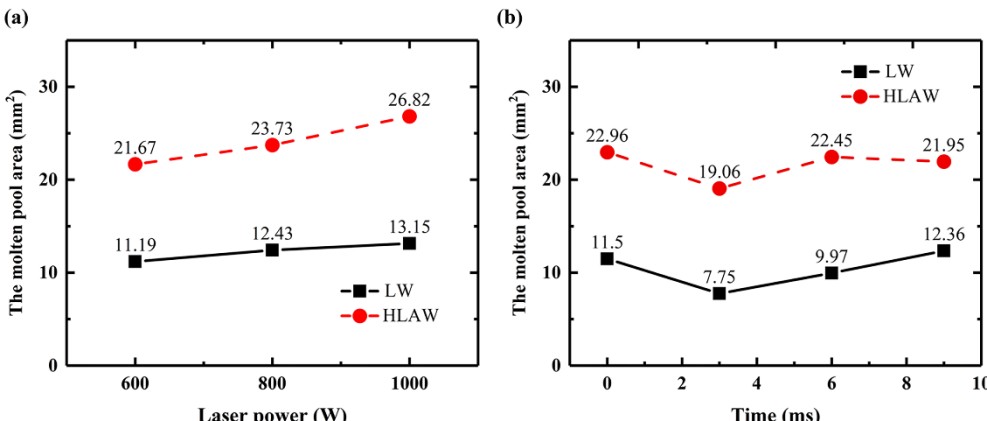

**Figure 5.** The molten pool area as functions of welding parameters: (**a**) laser power; (**b**) time in LW 600 and HLAW 600-20.

According to the above observations, the principal reduction of HLAW is shown in Figure 6. The molten pool fluctuation is caused by metal vapor recoil, gravity, and the surface tension of the molten pool [28]. Severe fluctuation of the molten pool results in pores forming, which reduces the welding quality to some extent in LW. However, because of the existence of pressure under the arc, HLAW alleviates the recoil force caused by metal vapor and inhibits the fluctuation in surface tension of the molten pool [29,30]. Consequently, the HLAW molten pool surface remains relatively stable. Furthermore, the larger arc thermal radius leads to an increase in the solidification time of the molten pool, which is conducive to the discharge of pores in the molten pool. These conditions improve the welding quality. Because of the increase in heat input, the molten pool area in HLAW is significantly larger than that in LW. Therefore, HLAW can be carried out at higher welding speed than LW, which improves the welding efficiency.

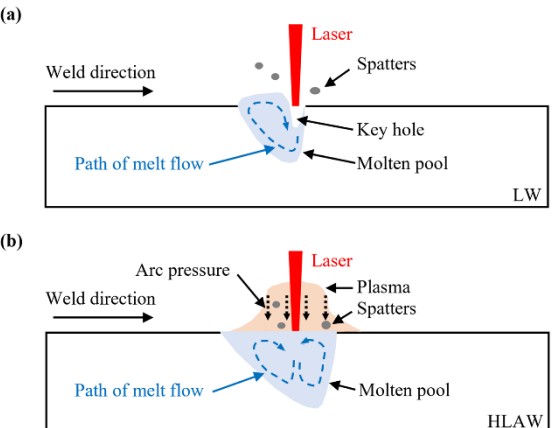

**Figure 6.** Schematic diagram of molten pool fluctuation mechanism: (**a**) LW; (**b**) HLAW.

*3.2. Microstructure*

Figure 7 shows the surface forming and cross section morphology weld seams with different welding modes. The front surface of all welded joints is basically well formed, and serious splashes of welded joints appear on the back surface. When the laser power is 600 W, the surface splash is the least in different welding modes. The energy input increases because of the increase in power. Thus, the molten pool reaction becomes more violent, and the spatters increase. On the other hand, the weld width of HLAW is significantly increased compared with LW. HLAW increases the heat input and the weld width accordingly. Serious undercuts were also observed in Figure 7b,c, which is unacceptable. This problem is mainly due to the high evaporation of the metal during welding, resulting in insufficient

material in the welding area. However, the undercut phenomenon of HLAW welded joints is significantly reduced compared with LW under each parameter in Figure 7d–f. This phenomenon is related to the arc pressure mentioned above, and HLAW suppresses the fluctuation in the surface tension of the molten pool.

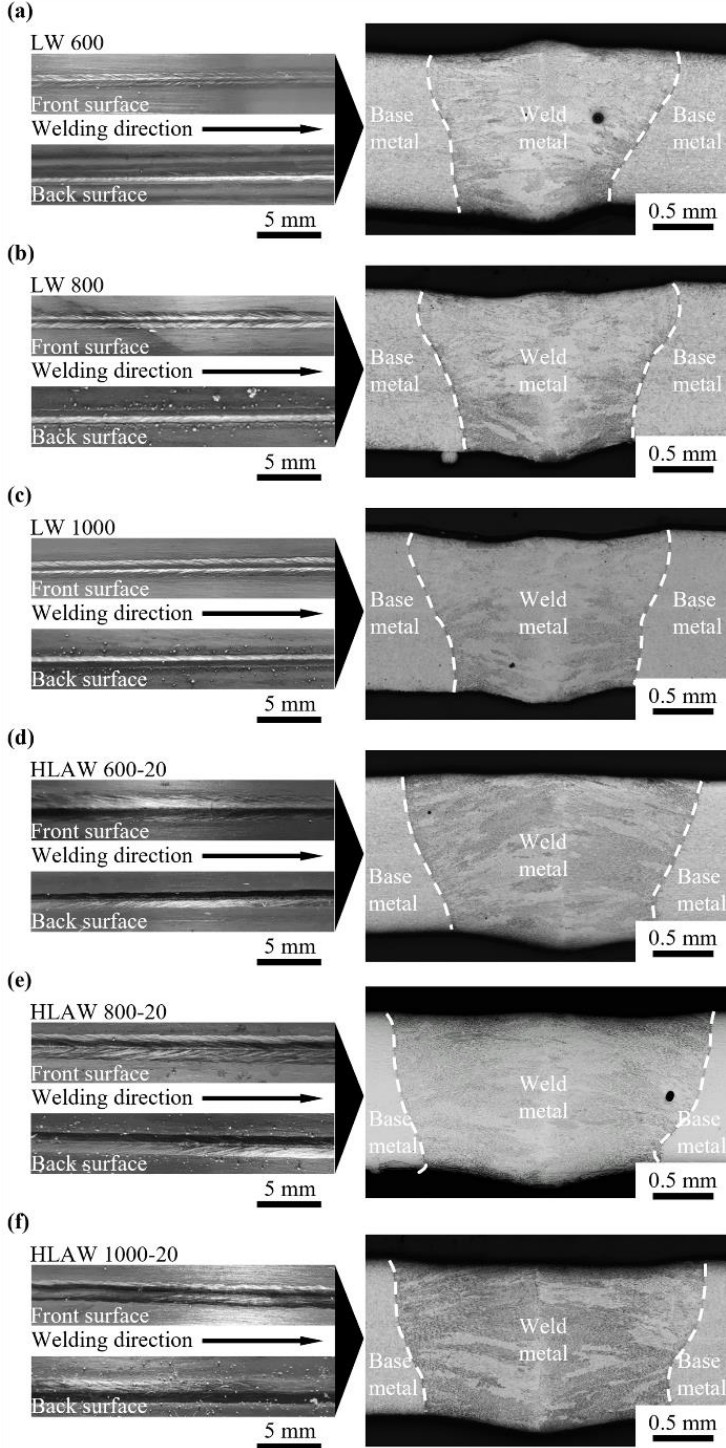

**Figure 7.** Surface forming and cross section morphology weld seams with different welding modes: (**a**–**c**) LW 600, LW 800, and LW 1000, respectively; (**d**–**f**) HLAW 600-20, HLAW 800-20, and HLAW 1000-20, respectively.

Figure 8 shows the microstructure of welded joint sections under LW and HLAW. It is obvious that a large amount of ferrite appears at all the welded joints. Ferrite precipitation

increases with uneven thickness in Figure 8a–c. Figure 8d–f show the microstructure of welded joint sections under HLAW 600-20, HLAW 800-20, and HLAW 1000-20 parameters, respectively. The ferrite is dense and uniformly distributed, which indicates that the joint microstructure under HLAW is more uniform than that under LW. The grain size of the welded joint was measured by Image J software, as shown in Figure 9. The crystal size of the weld structure of the HLAW welded joint is correspondingly reduced compared to LW at the same power. Undeniably, according to solidification theory, the lower the solidification rate in the same case, the larger the grain size should be, thus having higher hardness. However, the use of HLAW increases the electric field compared to LW, which will affect the formed size of the grain. Chen et al. [21] also found that the microstructure of 316L joints is mainly ferrite after HLAW. Shen et al. [31] found that with smaller heat input and formation of the heat affected zone, finer dendrite shapes were easily formed, which was beneficial to improve the mechanical properties of the formed layer. However, HLAW increases the arc heat input and reduces the heat source temperature gradient compared with LW. At the same time, owing to the larger radius of the area influenced by arc heating, HLAW increases the cooling time of the molten pool and reduces the cooling rate, which are not conducive to the precipitation of dendrites [21,32]. In other ways, HLAW increases the arc compared with LW, which will induce a directional electric field from the tungsten electrode to the substrate [33]. This is beneficial to the internal flow of the molten pool because of the directional current-generated magnetic field. Therefore, the joint microstructure under HLAW is more uniform than that under LW. Sabzi et al. [34] found that electromagnetic vibration can significantly reduce grain size and increase turbulence in the molten pool. Sabzi et al. [35] found that the microstructure of welded joints was refined under a pulse current mode. Under the pulse current mode, the heat input decreases and the temperature gradient decreases. The molten pool vibrates violently under the action of the pulse current, which makes the ferrite distribution more uniform.

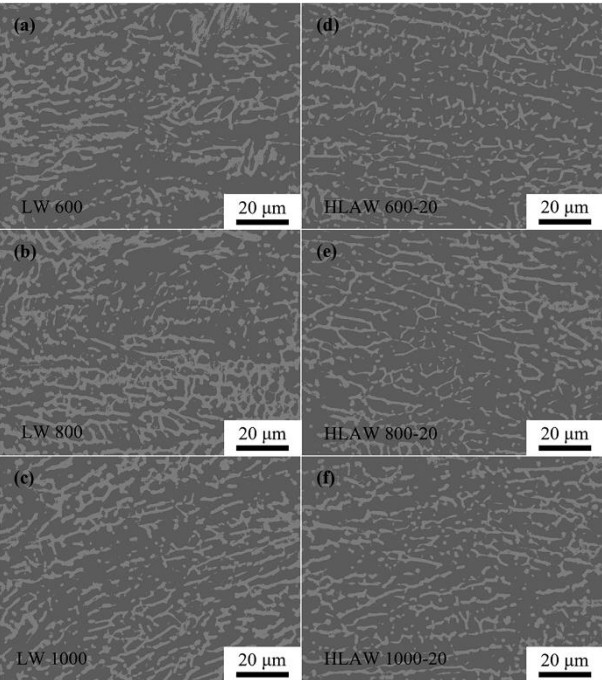

**Figure 8.** Microstructure image of the welding material obtained from an optical microscope and processed by image J: (**a–c**) LW 600, LW 800, and LW 1000, respectively; (**d–f**) HLAW 600-20, HLAW 800-20, and HLAW 1000-20, respectively.

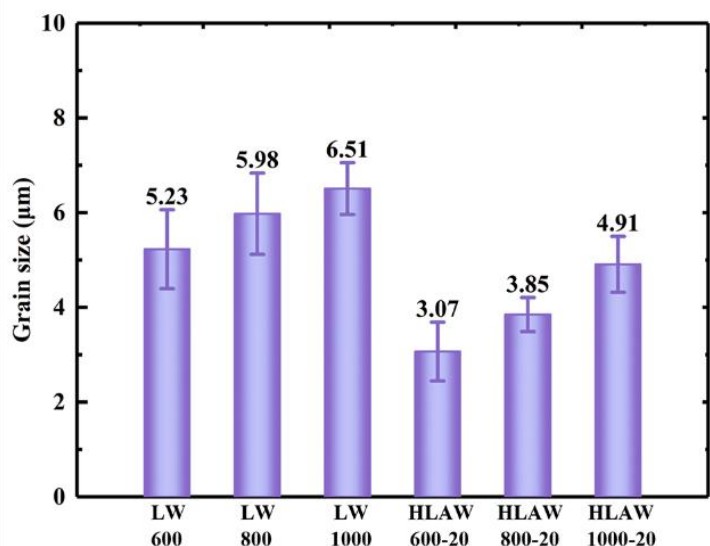

**Figure 9.** Grain size of welded joint under various parameters.

### 3.3. Microhardness

The microhardness of each specimen along the cross section was measured under 0.2-kg loading for 10 s. The microhardness results of the prepared samples are shown in Figure 10. The average microhardness of weld materials of LW600, LW800, and LW1000 is 188.6 $HV_{0.2}$, 189.3 $HV_{0.2}$, and 187.4 $HV_{0.2}$, respectively. The average microhardness of weld materials of HLAW600-20, HLAW800-20, and HLAW1000-20 is 195.4 $HV_{0.2}$, 191.1 $HV_{0.2}$, and 194.9 $HV_{0.2}$, respectively. It can be clearly seen that the microhardness of the sample joint is generally similar in each welding mode. However, the microhardness of weld materials of HLAW is significantly higher than that of LW. In Figure 9, the microstructure of weld materials of HLAW is finer than that of LW. At the same time, the refinement of ferrite helps to improve microhardness [36].

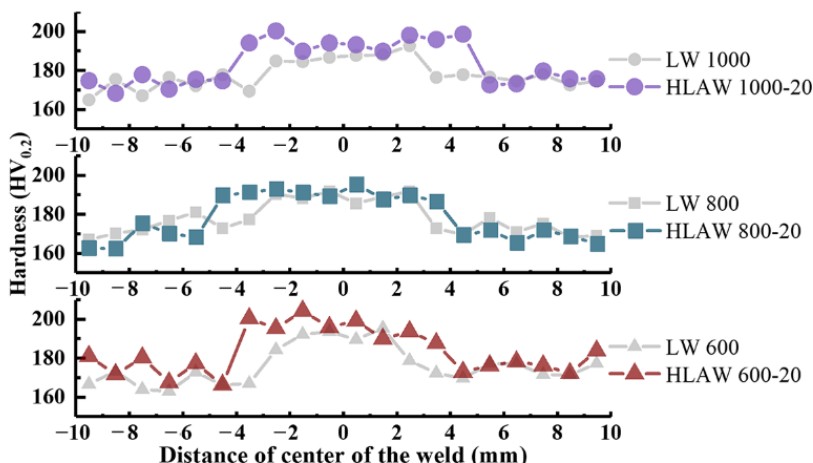

**Figure 10.** Microhardness distribution of weld section.

### 3.4. Tensile Strength

The tensile test results of the samples are shown in Figure 11. It can be seen from Figure 11a that the tensile strength of HLAW 600-20, HLAW 800-20, and HLAW 1000-20 specimens reached 544.7, 477.3, and 422.3 MPa, respectively. Compared with LW 600, LW 800, and LW 1000 samples, for which the tensile strength is 455.0, 439.7, and 417.3 MPa, respectively, the corresponding tensile strength increases successively. This indicates that HLAW is superior to LW in producing tensile mechanical properties of the specimens. The tensile strength of

LW 600, LW 800, and LW 1000 samples decreases in turn. The reason for this decrease may be that the increase in heat input is far away from the optimum process parameters. Because HLAW adds an arc on the basis of LW, this is the same as heat input. However, as shown in Figure 11, the tensile strength of the HLAW specimen is higher than that of LW. This further demonstrates that HLAW is superior to LW. Xie et al. [27] established the finite element model of HLAW and studied the influence of dynamic preheating on the thermal behavior of LW. They found that HLAW could significantly reduce the temperature gradient and cooling rate of the welded joints compared with LW, which would be beneficial to reduce thermal stress and improve the tensile properties of the welded joints. HLAW increases the arc heat source and optimizes the distribution of total heat source energy. This results in HLAW alleviating the rapid cooling of the LW. High temperature gradient will lead to high residual stress. The internal residual stress of welded joints is unfavorable to the tensile properties of joints. On the other hand, the ferrite distribution of HLAW welded joints is finer and more uniform than LW. Refined ferrite also helps to improve the mechanical properties of welded joints.

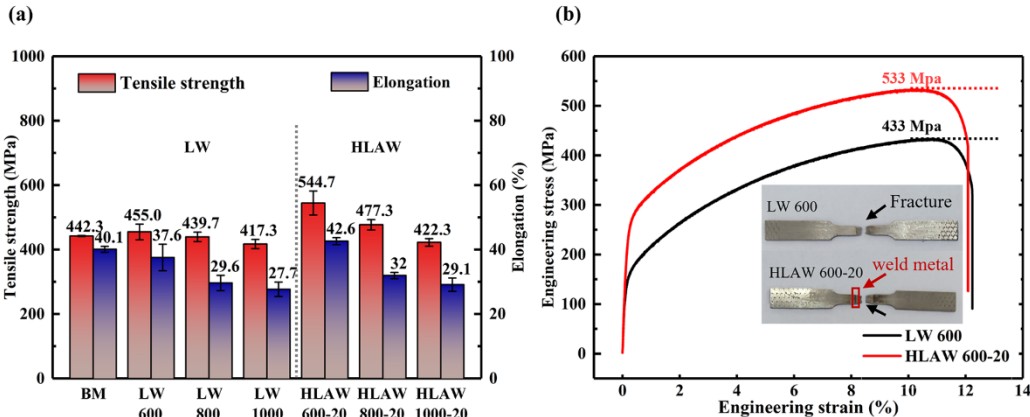

**Figure 11.** Tensile properties of different joints: (**a**) tensile strength and elongation; (**b**) σ-ε curves of the LW and HLAW joints.

The variation of elongation is similar to that of tensile strength. It can be seen from Figure 11a that the elongations of HLAW 600-20, HLAW 800-20, and HLAW 1000-20 samples reached 42.6%, 32%, and 29.1%, respectively, exhibiting a decreasing trend. Elongations of LW 600, LW 800, and LW 1000 samples were 37.6%, 29.6%, and 27.7%, respectively, and thus the corresponding elongations were improved under HLAW. Figure 11b shows the stress–strain curves under LW and HLAW to obtain the joints under the optimal parameters. The results show that the maximum tensile strength of the joint increases by 23.1% from 433 to 533 MPa.

Figure 12a,b show low-magnification fracture micrographs of LW 600 and HLAW 600-20 specimens, respectively. It can be seen from Figure 12a that there are many obvious defects in the fracture, which weakens the strength of the material. In Figure 12b, fracture defects are reduced. Figure 12c,d show enlarged graphs of the central region of Figure 12a,b, respectively. Figure 12c shows a low fracture roughness, and the fracture surface is mainly a tear ridge and an incomplete dimple. The deep dimple fracture in Figure 12d is obviously dense, which reflects a typical ductile fracture phenomenon. This indicates that plastic deformation occurs in the HLAW 600-20 specimen. It can be concluded that HLAW improves the tensile properties of joints.

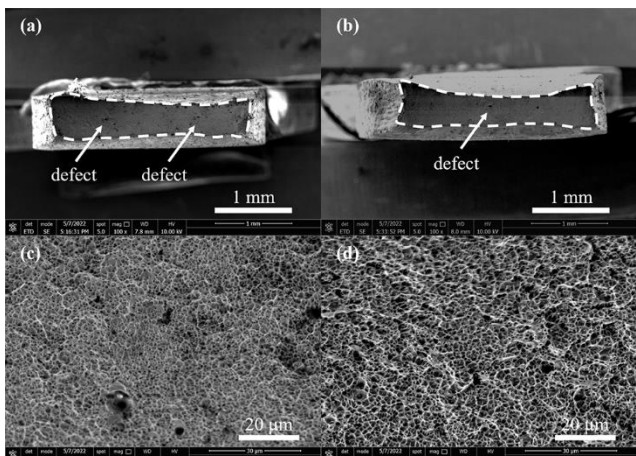

**Figure 12.** SEM of the fracture of the case: (**a**,**b**) LW 600 and HLAW 600-20, respectively; (**c**,**d**) higher magnification micrographs obtained from the Central areas in (**a**,**b**), respectively.

## 4. Conclusions

The following conclusions can be drawn from this study:

1. A comparison of LW and HLAW during the welding process by using a high-speed camera reveals that LW is more prone to molten pool splash than HLAW. The pool area of HLAW was significantly increased compared with that of LW. HLAW can also be carried out at a higher welding speed than that of LW to improve welding efficiency.

2. The microstructure of HLAW joints is more uniform and refined than that of LW joints, which can improve the microhardness and mechanical properties of welded joints.

3. HLAW improved the tensile properties of the joint, with the maximum tensile strength of the joint increasing by 23.1% (from 433 to 533 MPa).

**Author Contributions:** L.X.: conceptualization, methodology, formal analysis, writing—original draft, writing—review and editing. W.S.: conceptualization, formal analysis, supervision, project administration. T.W. and M.G.: discussions. D.C. and S.H.: equipment support. K.H.: fund. All authors have read and agreed to the published version of the manuscript.

**Funding:** This work was supported by the National Natural Science Foundation of China (Grant No. 62073089), the Foundation Project of Guangdong Province (2022A1515011118), and the GDAS' Project of Science and Technology Development (2022GDASZH-2022010203).

**Institutional Review Board Statement:** Not applicable.

**Informed Consent Statement:** Not applicable.

**Data Availability Statement:** Not applicable.

**Conflicts of Interest:** The authors declare that they have no known competing financial interests or personal relationships that could have appeared to influence the work reported in this paper.

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
