# Peer review of "Effect of Hybrid Laser Arc Welding on the Microstructure and Mechanical and Fracture Properties of 316L Sheet Welded Joints"

_metals, doi:10.3390/met12122181_

Round 1

Reviewer 1 Report (New Reviewer)

1.      In line 24, why have you introduced cheap stainless steel? What is your basis? Compared to what materials is it cheaper?

2.      In lines 42 and 43, why is the heat input lower in HLAW than in LW? All sources indicate that heat input is lower in LW.

3.      In line 122, isn't there a big gap between 0 ms and 9 ms? Has the boiling pool really been stable in all these cases?

4.      In the introduction, you mentioned that one of the advantages of LW is the smaller HAZ. Doesn't this area expand by adding an electric arc?

5.      You have proven that HLAW has less splash than LW. accept But you have mentioned (Fig. 5) that the area of the pond in HLAW conditions is twice that of LW. Does this not cause problems?

6.      Please explain in a suitable place why HLAW is better than LW? This can question the stability of input heat in LW, which is one of the most important advantages of lasers.

7.      On line 169 onwards you have indicated the larger dimensions of the weld pool in HLAW. Isn't that questioning the theme of your manuscript?

8.      In Figure 8, first, it seems that the images are electronic and not optical. And secondly, all the images are almost the same, which confuses the reader.

9.      Microhardness results section is not acceptable. Why is the hardness of HLAW samples higher than LW? Why are the grains of HLAW samples smaller than LW samples? In HLAW, the solidification speed is lower and this increases the grain size compared to LW. Therefore, it should have a higher hardness. What is the size of the grains in different parts of the weld?

Author Response

Reviewer 2 Report (New Reviewer)

Dear authors, thank you for the opportunity to review your work. The article deals with the study of the impact of the type of welding (laser beam welding or laser-arc hybrid welding) on the properties of welded 316L sheet joints. A deeper understanding of the interaction of welding parameters with 316L steel will allow a more efficient use of this material in the specific industrial applications. For this reason, I find the subject of your work quite important. Your manuscript is well structured. The graphic material is of good quality and self-explanatory.

Nonetheless, I have some comments on the manuscript:

- in particular, I notice that the keywords do not adequately reflect the content and main findings of the work. Please revise and complete the related keywords.

- in the introduction section, you rightly highlighted the advantages of laser hybrid welding over laser welding. However, it must be taken into account that there are several types of laser hybrid welding, depending on the arc being use:. TIG, plasma arc or MIG augmented laser welding. Also, laser hybrid welding often involves the use of filler material in the form of wire or metall powder. These aspects are not covered in the introduction, and it is unclear what type of laser hybrid welding is considered in your paper. Please revise the introduction in this context and indicate that you are working with a specific type of laser hybrid welding (laser-GTAW hybrid welding).

- 3.4. Tensile Strength: In this section, you mention the term yield strength in the text. As far as I know, the yield strength cannot be determined on welded specimens because the material is not homogeneous. There are areas such as the base material, the heat-affected zone and the weld metal that have different properties. In this case, the term tensile strength is more meaningful, as you apply it to the description of Fig. 10. Please consider these aspects and revise the section 3.4.

- please indicate whether the fracture occurred through the weld metal or through the base metal or HAZ. 

- plese set a spase before brakets []. Check punctuation throughout the text.

Round 2

Reviewer 1 Report (New Reviewer)

The authors have answered all questions well and clearly except one. The authors have said “Undeniably, according to solidification theory, the lower the solidification rate in the same case, the larger the grain size should be, thus having higher hardness” in response to question 9.  This is not acceptable at all. As grain size increases, hardness decreases rather than increases. It is true that in HLAW ferrite refinement occurs and hardness increases. But this is not enough. The role of grain size is also important. After this matter becomes clear, it is recommended to publish the manuscript.

Author Response

Reviewer 2 Report (New Reviewer)

Dear Authors, I have one more remark about part 3.4. In my previous review comments, I meant that yield strength is not applicable for welded joints because the tensile test specimen has regions with inhomogeneous properties. As far as I know, it is not practicable to determine the yield strength at the welds. However, this is possible for the characterization of the pure weld metal or base material, since the plaso-elastic properties of the test specimen (material) are homogeneous. Therefore, I think the term "tensile strength" or "ultimate strength" is more accurate when testing welded joints. In this regard please refer to the technical literature, for example ISO 4136 - Destructive tests on welds in metallic materials and correct the terminology in the manuscript.

Author Response

This manuscript is a resubmission of an earlier submission. The following is a list of the peer review reports and author responses from that submission.

Round 1

Reviewer 1 Report

1.      Laser welding has advantages in the field of stainless-steel sheets welding. when the welding parameters are reasonable, the qualified weld formation and joint properties can be obtained. The relevant research in the manuscript is insufficient. Is it necessary to apply HLAW for stainless steel sheet?

2.      Authors point out that LW leads to hardening of the microstructure of the weld. However, the Fig. 6 shows that the microhardness of HLAW welded metal is higher. Especially for the HLAW 600-20, the hardening of weld metal is severe. In addition, it seems that there is no difference of weld width and HAZ width between LW and HLAW. It is not reasonable.

3.      When the laser is inside the TIG nozzle, how is the arc stability? It is suggested to provide arc voltage and current waveform. In addition, the photo of weld surface and cross section should be added.

4.      In figure 3, it seems that the molten metal flows downward in HLAW, and there are spatters near the molten pool. From the position of the arc and laser, the red line only marks the end of the weld pool, not the complete weld pool.  For stainless steel sheet, spatters are more likely to form at the front of molten pool or at the back of the sheet. The comparison of papers is not targeted.

5.      Where are the fracture positions of tensile specimens? What is the strength of the base metal?

6.      The defects in the fracture, which weakens the strength of the material, are pores? For stainless steel, nitrogen shielding is helpful to reduce porosity.

Reviewer 2 Report

This paper discusses the effect of hybrid laser arc welding on the performance of the most popular stainless steel(316L). The equipment and procedures used in the experimental process were described in detail. The effects of laser welding and hybrid laser arc welding on the splash behaviours of the molten pool, microstructure, microhardness, tensile strength and fracture morphology were compared and analysed. However, there are still some aspects that need to be improved.

1.   In the introduction, please increase the 316L stainless steel related material knowledge introduction.

2.   Please add the novelty of the paper to the introduction.

3.   In Figure 2b, please mention the standard for tensile specimen.

4.   In Figure 6, why is there less hardness in some samples in the HAZ area than in the BM? Please give the corresponding explanation.

5.   In Figure 7a, the tensile strength was not tested on the 316L base material, please supplement the corresponding data.

6.    The results are good, but further discussion is needed to link the results of the various parts.

Reviewer 3 Report

In this study, welding tests were conducted on 316L thin-plate using a laser arc hybrid gun head similar to a coaxial type. A high-speed camera was used to record the welding process and analyze the droplet splash behavior of the molten pool. The microstructure, microhardness change, and tensile test results of welded joints under different welding modes were analyzed.

This paper mainly focuses on the description of the phenomenon but lacks in-depth analysis, which is inconsistent with the research field and direction required by the journal. Especially the part about microstructure analysis and mechanical property analysis lacks convincing direct evidence. It is suggested to modifed this part of the content or transfer to other journals.

Round 2

Reviewer 1 Report

The manuscript has not been sificiently improved. The supplement is not enough to demonstrate the novelty of the proposed method. 

What problem does the proposed method solve in the welding of thin stainless steel? Compared with separating heat sources, what are the advantages of putting arc and laser in one welding torch? There is no clear answer in the revised version.

In the response, authers point  out that "All tensile specimens were fractured in the middle of the welded joint. " Actually the strength of HLAW specimen is much higher than base metal.  Why does the fracture not occur at the weak base metal? 

Authors are unable to provide supplementary data on weld photos and welding process signals.  This makes the repeatability of the method untrustworthy.

There is no information about the effect of porosity on fracture strength in the author's response.

Reviewer 3 Report

The paper entitled "Effect of Hybrid Laser Arc Welding on the Microstructure and Mechanical and Fracture Properties of 316L Sheet Welded Joints" and the test method used is innovative. However, there are significant problems in the grammar of the paper and the usage of critical proper nouns. It is recommended that a native speaker be invited to make major revisions to the entire paper. In addition, many critical conclusions of the paper lack direct basis and evidence. Therefore, this paper fails to meet the requirements of journal publication, and it is recommended to reject the manuscript.
